# Effect of Ornithine Transcarbamylase (OTC) Deficiency on Pregnancy and Puerperium

**DOI:** 10.3390/diagnostics12020415

**Published:** 2022-02-05

**Authors:** Rastislav Sysák, Katarína Brennerová, Romana Krlín, Peter Štencl, Igor Rusňák, Mária Vargová

**Affiliations:** 11st Department of Gynaecology and Obstetrics, Faculty of Medicine, Comenius University in Bratislava, 851 07 Bratislava, Slovakia; stenclp@centrum.sk (P.Š.); maria.vargova@faltus.sk (M.V.); 2Department of Pediatrics, National Institute of Children’s Diseases, 831 01 Bratislava, Slovakia; dfn.fabriciova@gmail.com; 32nd Department of Gynaecology and Obstetrics, Faculty of Medicine, Comenius University in Bratislava, 821 01 Bratislava, Slovakia; romikrlin@gmail.com; 41st Department of Gynaecology and Obstetrics, Faculty of Medicine, Slovak Medical University, 831 01 Bratislava, Slovakia; irusnak@pobox.sk

**Keywords:** ornithine transcarbamylase deficiency in pregnancy, puerperium, hyperammonemia, hepatic failure

## Abstract

Ornithine transcarbamylase (OTC) deficiency is the most common inherited metabolic disorder in urea cycles with an incidence of 1:14,000 live births. Pregnancy, childbirth and the postpartum period are considered challenging for women with this hereditary metabolic disorder, with a risk of hyperammonemia, especially in the first week after delivery. In our article, we discuss severe hepatic failure, a pregnancy complication in an OTC deficient patient that has not previously been published. Firstly, our aim is to highlight the need for a strict adherence to the recommendation of the gradual increase of protein intake during pregnancy and the importance of multidisciplinary monitoring of pregnant patients with OTC deficiency. Secondly, we refer to critical postpartum hyperammonemia in patients with this hereditary metabolic disorder.

## 1. Introduction

Ornithine transcarbamylase (OTC) is an enzyme in the urea cycle, located in mitochondria. It catalyzes the reaction between carbamoyl phosphate and ornithine to form citrulline [1]. OTC deficiency is the most common inherited metabolic disorder in the urea cycle [2,3]. The disease is bound to the X chromosome. In male individuals, hemyzygotes, the disease often manifests as a severe hyperammonemic crisis with a neurological symptomatology as early as in the neonatal period [4]. If not recognized correctly and left without treatment, the disease can progress to severe brain edema and lead to the death of the newborn [2]. Heterozygous women may be asymptomatic or with varying degree(s) of clinical manifestation. The degree of severity of the disease is quite variable in women and depends on the degree of inactivation of the mutated X-chromosome [5]. In mild courses, attacks are often present after ingestion of a high-protein meal [6], subconsciously leading to a reduction of protein in their diet. This sometimes delays diagnosis of OTC deficiency. More severe enzyme deficiency causes symptoms as early as in early childhood—recurrent attacks of vomiting, growth retardation, hypotension, psychomotor retardation, hyperammonemic coma or psychiatric difficulties. OTC deficiency diagnosis is determined by hyperammonemia with orotic aciduria, citrulline deficiency and elevation of glutamine levels in the plasma amino acid profile. In the pedigree of these patients, it is possible to detect an unexplained death of newborn males due to X-linked inheritance. In the past, the diagnosis was made by examining the activity of the enzyme in liver or intestine cells from biopsy samples. Currently, molecular genetic analysis of the OTC gene is the method of choice to confirm the diagnosis [7]. This is also used for prenatal diagnostic purposes. The basis of treatment is a low-protein diet, according to individual tolerance, pharmaco-elimination therapy (sodium benzoate, phenylbutyrate) and L-arginine with L-citrulline supplementation, whose blood levels decrease due to urea cycle dysfunction. Severe forms of the disease are an indication for a liver transplant [8]. In early childhood, it is possible to use a hepatocyte infusion through a portal vein as an alternative to liver transplant, or as an abridging therapy to a subsequent liver transplant [9]. Pregnancy can be a high-risk condition in patients with OTC deficiency, as catabolic processes predominate, especially in the intrapartum and postpartum periods. These processes can lead to life-threatening hyperammonemic encephalopathy and coma. The period between the third and the fourteenth day after delivery is considered to be the most crucial for the occurrence of metabolic decompensations, although it can also occur 6 to 8 weeks after delivery. In pregnant patients with OTC deficiency, the most important preventive measures during this period are a significant reduction in protein intake and sufficient energy supply.

Once unexpected neurological deterioration of the patient in puerperal period occurs, the obstetrician should be aware of possible unrecognized metabolic disease, e.g., OTC deficiency.

Several publications describe postpartum complications in OTC deficient patients [10]. In our case report, we describe a serious complication manifesting in the second trimester of pregnancy in a 21-year-old woman with OTC deficiency. We would like to draw attention to the importance of a strict adherence to treatment measures, not only in the postpartum period, but throughout the whole pregnancy in patients with OTC deficiency.

## 2. Case Description

21-year-old primigravida with a proven heterozygous mutation in OTC gene, treated from her 9 months of age. During childhood, the patient was often hospitalized for metabolic decompensations mostly caused by insufficient energy intake. Our patient and her partner were properly counselled about the course of OTC deficiency disease and its type of genetic inheritance, as well as about the possibilities of prenatal diagnosis, which was not agreed on by the partners.

Prior to pregnancy, her long-term treatment included a low-protein diet containing natural proteins of 29 g per day, sodium-benzoate and sodium-phenylbutyrate four times per day, 4 g and 2 g, respectively, and supplementation of L-arginine, L-citrulline, L-carnitine, calcium and group B vitamins.

Essential amino acids supplementation was not taken as prescribed, as the patient did not tolerate them well. Reportedly, the patient suffered nausea and vomiting after essential amino acid administration. The patient visited metabolic and pregnancy counselling regularly from the beginning of her pregnancy. In the first trimester, the patient was hospitalized for morning sickness. During this period, treatment with sodium phenylbutyrate was discontinued for its potential teratogenicity [11].

In the first trimester, fetus development and patient metabolic compensation were adequate. At the 22nd week of pregnancy, the patient was re-hospitalized for vomiting and a risk of metabolic decompensation. A plasma ammonia level of 135.5 μmol/L (reference values 0–60 μmol/L) was gradually optimized with glucose infusion administration, however plasma albumin levels decreased to 18.4 g/L (reference value 35 to 50g/L) and coagulation parameters deteriorated with an INR value of 2.13 (normal value up to 1.3). The patient did not tolerate enteral intake sufficiently, so she required high-energy parenteral nutrition through a central venous catheter, frequent albumin replacement and coagulation status adjustment. We initiated a long-term intravenous sodium-benzoate and arginine administration. Despite fortified treatment, an increase in ammonia concentration to the level of 221 μmol/L was reported. Due to impaired consciousness, the patient was transferred to the metabolic ICU ward at the 25th gestational week, where the complete differential diagnosis of hepatopathy was realized. Neither viral nor autoimmune etiology has been confirmed. Sodium-phenylbutyrate treatment was reinitiated. Within 5 days, the patient’s condition stabilized. Plasma ammonia levels decreased to normal. There was a slight increase of plasma albumin and cholinesterase levels and INR dropped to 1.36. The patient was discharged from ICU and readmitted to a regular ward of the internal medicine department in order to continue treatment and monitoring of metabolic compensation (Figure 1).

Due to the previous intolerance of oral amino acids intake, we added a small amount of intravenous amino acids substance to the regular intravenous therapy and albumin substitution.

Within the 28th week of pregnancy, the patient started to vomit again. There were repeated reports of an increase in the plasma ammonia level with a maximum of 176 μmol/L as well as a decrease of INR value to 1.7. At the same time, the patient reported transient weak fetal movements.

Given that it was the 27th⁺⁵ week of pregnancy, after considering all risks and benefits, the patient was transferred into the perinatal care center and antenatal corticosteroid administration for fetal lung maturation was initiated. Afterwards, multidisciplinary assessment with the participation of a gastroenterologist, perinatologist, hematologist, anesthesiologist, internal medicine, and metabolic disease physician was performed with the resolution to prepare the mother for delivery by cesarean section. Right before surgery, we administered clotting factor concentrate (clotting factor II, VII, IX and X) to the patient at a dose of 25 IU/kg. The cesarean section was carried out at the 28th⁺¹ week of pregnancy under general anesthesia. A feminine neonate of 870 g was born with an Apgar score of 4/6/7. The surgery outcome was without complications, with adequate perioperative blood loss with up to 500 mL. A Redon drain was inserted into the subfascial space to control potential subsequent bleeding. Due to the underlying disease with a risk of hepatic damage related to a high risk of possible postoperative complications (hyperammonemia, hepatic failure and consciousness disturbances) the patient care was managed in the ICU ward following the cesarean section. In the early postoperative period, daily coagulation and biochemical parameters were monitored. Coagulation parameters normalized within 2 days after the C-section without the need for further substitution. The plasma ammonia level was normal during the first three days after delivery. From day 4 after surgery, there was a gradual increase in ammonia levels with a peak of 91 μmol/l (reference value ˂35 μmol/L). After dietary proteins intake reduction and sodium-benzoate dose adjustment to the original dosing of 4g four times per day, the plasma ammonia level normalized. Having a generally favorable postoperative course, the patient was transferred to the internal medicine department on the 12th day after C-section. Due to lactation, it was necessary to re-increase protein intake to 40 g/day. The patient was discharged home 4 weeks after delivery with improving liver function tests, without coagulopathy.

Plasma ammonia levels was in the physiological range of 17–30 μmol/L. Cholinesterase and plasma albumin levels normalized 7 weeks and 9 weeks after delivery, respectively (Figure 2).

At the end of the first pregnancy the patient was considering undergoing sterilization during the planned C-section, thus, due to the unexpected preterm delivery in the 28th week of gestation, the administrative requirements for conducting a sterilization procedure were not met. For the reason of a potential risk of worsening hepatic functions, the hormonal contraception was not advised. We discussed the use of barrier contraceptive methods with the patient and her partner.

Currently—6 years after birth—the child did not show any signs of possible negative effects of hyperammonemia of the mother during pregnancy. At the same time, no medical conditions due to severe prematurity were reported. Nowadays, the girl attends the 1st year of primary school.

Four years after the first pregnancy, the patient became pregnant again. From the beginning of her second pregnancy, we consistently monitored the protein intake with its gradual increase as recommended (Table 1 [8]). We did not discontinue glycerol phenylbutyrate treatment. Throughout the pregnancy, the cholinesterase level was in a normal range up to the eight month of gestation, when it transiently decreased slightly (Figure 3). There were no signs of coagulopathy, and the INR value was satisfactory.

Despite the gradual increasing of protein intake, the ammonia level was kept below 80 umol/L, with only isolated fluctuations associated with insufficient energy income (Figure 4). At the end of this second uncomplicated pregnancy, a planned C-section at the 36th⁺⁴ gestational weeks was performed. Again, the feminine neonate with a fetal birth weight of 2700 g and a fetal length of 48 cm was born. In the first week of puerperium, there was a significant reduction of protein consumption and increased caloric intake. Otherwise, the puerperium period passed without complications.

Although the second conception was not the result of a precise pregnancy planning, the course of this pregnancy was uncomplicated and without any unpredicted situations. Following the required policy, we performed the tubal ligation during the second cesarean section.

## 3. Discussion

Ornithine transcarbamylase deficiency is the most common hereditary metabolic disorder in the urea cycle. Symptoms of this disorder in women with mild enzyme deficiency are not specific. Frequently, these are migraine headaches, recurring nausea and vomiting, lethargy, hyperventilation, behavior changes, mental confusion, ataxia and anorexia mostly manifested after ingestion of food with an increased protein content [12,13]. Episodes of acute metabolic decompensation may lead to a life-threatening hyperammonemic encephalopathy. The basis for treatment of ornithine transcarbamylase deficiency is a low-protein diet and appropriate pharmaco-elimination therapy, as well as L-arginine and L-citrulline substitution and a reduction of trigger factors of metabolic decompensation, such as febrile diseases or gastroenteritis [3].

Pregnancy, and puerperium in particular, is one of the main medical conditions requiring a management modification and multidisciplinary cooperation [10,12,13,14,15,16]. Mostly anorexia and nausea in the first trimester put the patient at risk of a hyperammonemic crisis. For this reason, it is crucial to ensure a sufficient energy supply for the pregnant woman and prevent dehydration. The low-protein diet is regulated during pregnancy due to anabolic demands of the fetus, and a gradual increase in protein intake is recommended. In the third trimester, the increase is up to 31 g per day more than the protein intake in the pre-pregnancy diet [8]. For dietary restrictions, it is important to regularly monitor the growth and development of the fetus. In most published cases, this growth and development was not complicated [14]. In our case report, the fetal biometric parameters in the 28th⁺¹ gestational week corresponded to the 26th week of pregnancy, which we considered potential fetal hypotrophy.

Vaginal delivery is associated with an increased energy demand and the risk of dehydration given that it is important to prevent protracted labor and provide sufficient hydration with adequate caloric supply to a patient.

Early epidural analgesia administration is appropriate to reduce the catabolic mechanisms followed by endocrinological response to stress [12]. In patients with OTC deficiency, a programmed labor in tertiary care is favored [13].

Thus far, there is no clear explanation as to why the early puerperium is a high-risk period for the patient. Hypothetically, the combination of the termination of nitrates consumption by the fetoplacental unit and uterus involution, results in increased patient’s plasma protein level followed by hyperammonemia [14].

According to Lamb and co. [17], there were only 20 cases of ornithine transcarbamylase deficiency in pregnancy published until the year of 2013 [12,13,14,15,16,17,18,19,20]. Of these, there were 18 live born neonates reported. In 12 of these published cases, the diagnosis of deficit ornithine transcarbamylase was established prior to pregnancy. Six out of 20 patients had an uncomplicated course of pregnancy and puerperium, and they delivered in term. Thorough management of these patients contributed to an uncomplicated course of pregnancy [13,16]. In other cases, the major cause of complications was the hyperammonemic crisis [12,14,15,18].

Reportedly, eight patients were diagnosed with ornithine transcarbamylase deficiency during the pregnancy on the basis of differential diagnosis of hyperammonemia, which led to brain edema and subsequent death in five of these patients [15,18,20].

Swiss authors studied the incidence of acute liver failure in patients with OTC deficiency. A higher incidence of acute liver failure was detected in male patients. In women with a mild course of the disease, this complication did not occur [21]. The authors did not indicate if there was acute liver failure during pregnancy.

Acute liver failure as part of the presentation of undiagnosed OTC deficiency at the beginning of the pregnancy in one patient was described by Weiss N et al. [22]. To date, in our metabolic diseases center, two pregnant women with OTC deficiency were managed. Both of them subsequently became pregnant multiple times. The first patient rejected regular care in the metabolic center. In this patient, there were three uncomplicated courses of pregnancy from a metabolic perspective. However, in the first two early puerperal periods, signs of hyperammonia were reported on day 5 after delivery with plasma ammonia levels of 236.7 μmol/L and 250 μmol/L, respectively. A failure to comply with the recommended postpartum dietary management by the patient was attributed as the cause.

For the twenty-one-year-old patient presented in our case, no signs of acute liver failure were reported during her life-time medical follow-ups, despite a number of severe metabolic crises with hyperammonemia. From the beginning of her first pregnancy, she was regularly checked by a metabologist and a gynecologist. Delivery by cesarean section was planned at a perinatology center adjacent to the metabolic center. The second half of the pregnancy was complicated by synthetic liver function disorder with intermittent hyperammonemic crises which seriously endangered both the mother and the fetus. For this reason, acute management of this condition was necessary. We decided to deliver the fetus by cesarean section in the 28th gestational week from the vital indication of the mother. She gave birth to a hypotrophic but otherwise healthy girl who was not diagnosed by OTC gene mutation.

After delivery, the patient’s liver function recovered rapidly, and parenteral therapy could be discontinued 4 weeks after C-section. Our thorough retrospective reassessment of this pregnancy resulted in the implication that the serious synthetic liver function disorder might be caused by an insufficient protein intake in the second trimester [8]. The significant liver load after the discontinuation of sodium-phenylbutyrate at the beginning of pregnancy cannot be excluded. This assumption is also supported by a rapid improvement of synthetic liver function after delivery, and especially by the uncomplicated course of the patient’s second pregnancy.

## 4. Conclusions

In our case, we aimed to point out the unusual pregnancy complications of a patient with OTC deficiency. Seriously impaired synthetic liver function required multidisciplinary access. A proper therapeutic management was reached to maintain the pregnancy until fetal viability.

Preterm delivery by cesarean section was performed from the vital indications of the mother and to avoid toxic effects of hyperammonemia on the fetus.

In patients with ornithine transcarbamylase deficiency, special attention needs to be paid to preconception consulting on lifestyle and potential treatment in pregnancy with a focus on the necessary adjustment of the protein-boosting diet. This may be challenging in women who have followed a strict low-protein diet before pregnancy for an extended period of time, particularly in women with more serious forms of OTC deficiency. Contrary to that, a protein reduction with otherwise increased caloric intake is crucial in the first 2 weeks after delivery as catabolic processes predominate.

## Figures and Tables

**Figure 1 diagnostics-12-00415-f001:**
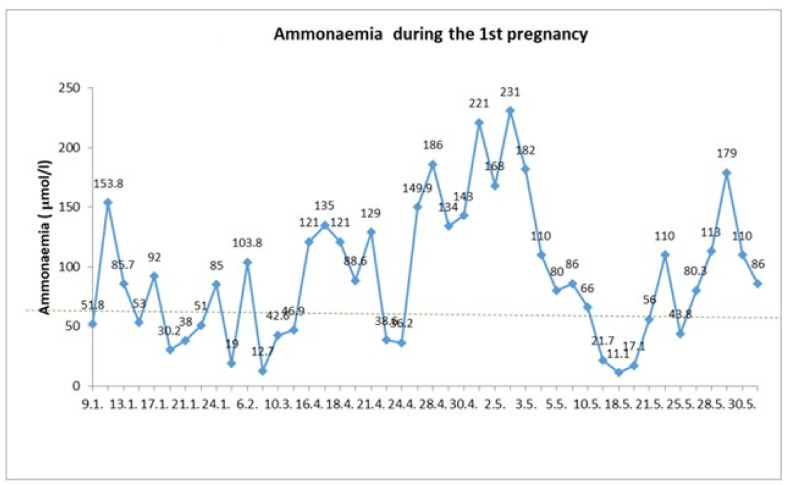
The course of ammonemia during the first pregnancy. The patient was hospitalized in multiple specialized clinical settings from 16 April until 1 June 2015. Despite the high-energy intake managed by the administration of 20 and 40% glucose infusions and intravenous arginine and sodium-benzoate, the effort to reduce ammonia levels to the normal values (below 60 μmol/L) did not succeed for an extended time. Signs of impaired consciousness on May 2nd and an ammonia level rise to 231 μmol/L resulted in the reinitiation of sodium-phenylbutyrate treatment, which had a transient positive effect on the plasma ammonia level. Despite an unchanged treatment, on May 29th, the ammonia level rose to 179 μmol/L again and the patient reported transient weak fetal movements. Initiation of corticosteroid fetal lung maturation was indicated and on 1 June delivery by C-section was performed due to the potential risk for the mother and her fetus.

**Figure 2 diagnostics-12-00415-f002:**
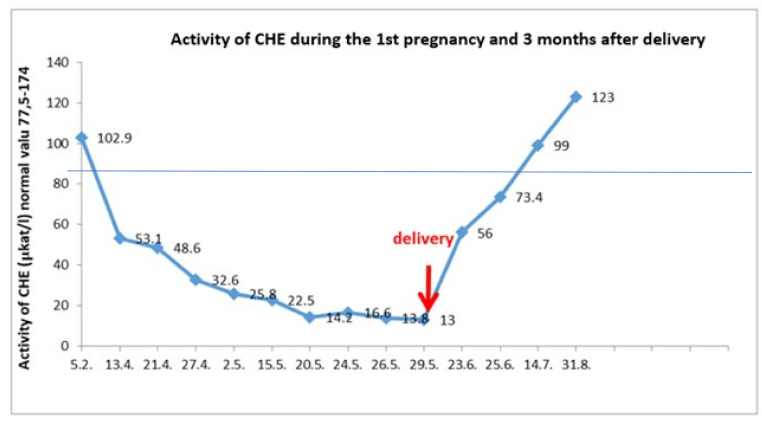
Cholinesterase plasma values during the first pregnancy. At the prenatal visit in April 2015, a decreased albumin level was recorded. Since then, CHE level and other synthetic liver function indicators had been progressively decreasing, coagulation factor V in particular. After delivery, these parameters were spontaneously improving without a treatment modification, and after seven weeks they were completely normal.

**Figure 3 diagnostics-12-00415-f003:**
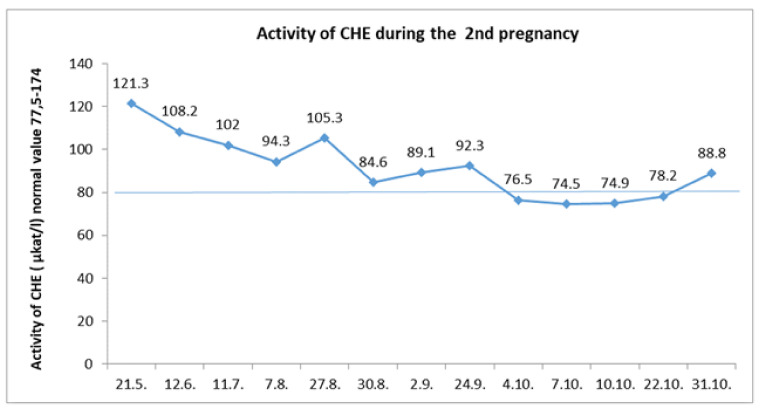
During the second pregnancy, cholinesterase levels were decreasing physiologically, especially during the first trimester. A temporary decrease below the normal values was reported in the third trimester, which recovered spontaneously.

**Figure 4 diagnostics-12-00415-f004:**
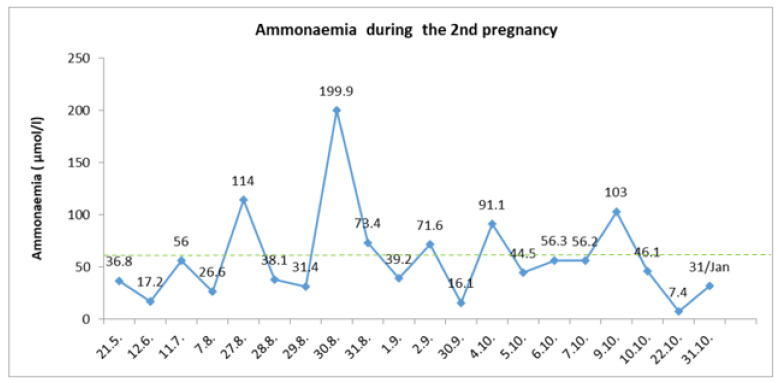
Ammonia levels during the second pregnancy were more stable. The patient was hospitalized for the first time at the end of the 2nd trimester from 27 August to 2 September for signs of fatigue and nausea. Intravenous energy support and diet adjustment were required. In October 2019, the patient was hospitalized for a perinatal management assessment when she missed her treatment. This was followed by metabolic decompensation and hospitalization was necessary for the need of infusion therapy.

**Table 1 diagnostics-12-00415-t001:** Recommended amounts of gradual increase of energy and protein intake during pregnancy compared to a regular diet of patients with OTC deficiency (Haberle 2012).

	Daily Energy Intake Increase	Daily Protein Intake Increase
trimester	kcal/day	grams/day
1	+90	+1
2	+287	+10
3	+466	+31

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
