# Peer review of "Effect of Ornithine Transcarbamylase (OTC) Deficiency on Pregnancy and Puerperium"

_diagnostics, 2022, doi:10.3390/diagnostics12020415_

Round 1
Reviewer 1 Report
The case study described in the article shows the course of two pregnancies in one woman with a genetic metabolic disorder - ornithine transcarbamylase (OTC) deficiency. For myself, I recommend publishing the article in this form
Author Response
Thank you for giving us the opportunity to submit a revised draft of the manuscript “Effect of ornithine transcarbamylase (OTC) deficiency on pregnancy and puerperium” for publication in the Diagnostics. We appreciate the time and effort that you and the reviewers dedicated to providing feedback on our manuscript and are grateful for the insightful comments and suggestions for our paper. We have incorporated most of the suggestions made by the reviewers. Those changes are highlighted within the manuscript. At the same time, we considered useful to add a cut-off line of normal values in figure 2 a figure 3. Please see below, for a point-by-point response to the reviewers’ comments and concerns. All page numbers refer to the revised manuscript file with tracked changes.
Reviewer I: The case study described in the article shows the course of two pregnancies in one woman with a genetic metabolic disorder - ornithine transcarbamylase (OTC) deficiency. For myself, I recommend publishing the article in this form
Author response: Thank you!
Reviewer 2 Report
I have no suggestions
Author Response
We appreciate the reviewer’s brief feedback in the general evaluation. Our manuscript was revised and updated in every section to approach the reviewers’ expectations and provide a better understanding of our OTC deficiency management in pregnancy and puerperium. Please find the updates in the revised manuscript highlighted with the tracked changes. Thank you very much for your expert opinion!
Reviewer 3 Report
The case report is written well in detail.
It would be appropriate to add whether pre-pregnancy advice was given prior to starting pregnancy.
Further, please add post-partum advice on contraception as well as advice on future pregnancy plan.
Author Response
Thank you for your expert opinion and suggestions.
Point 1: It would be appropriate to add whether pre-pregnancy advice was given prior to starting pregnancy.
Author response. Thank you for pointing this out. Our patient and her partner were properly counselled about the course of OTC deficiency disease and its type of genetic inheritance as well as about the possibilities of prenatal diagnosis. The revised text reads as follows on the page No.2, line 82-85.
Point 2: Further, please add post-partum advice on contraception as well as advice on future pregnancy plan.
Author response to point 2: We think this is an excellent suggestion. Therefore, we have addressed this concern and have added it into the manuscript. This change can be found on page 3-4, paragraph 153-158 and on page 4, paragraph 177-180.
Reviewer 4 Report
The authors discussed hepatic failure which in an unpublished OTC deficient patient, and found importance of strict adherence to the recommendation of the gradual increase of protein intake during pregnancy and multidisciplinary monitoring of patient with OTC deficiency. Additionally, authors refer to critical postpartum hyperammonemia in patients with OTC. This finding is novel, but more cases are necessary to support hypothesis.
Author Response
Thank you for the reviewer’s evaluation. Our manuscript was revised and updated in every section to approach the reviewers’ expectations and provide a better understanding of our OTC deficiency management in pregnancy and puerperium. Please find the updates in the revised manuscript highlighted with the tracked changes. We agree with the reviewer’s statement that more publications and discussions are needed to choose the best available management of the OTC deficiency diagnosis in a pregnancy and post-partum.